# Performance Analysis of Colored PLA Products with a Fused Filament Fabrication Process

**DOI:** 10.3390/polym11121984

**Published:** 2019-12-02

**Authors:** Roberto Spina

**Affiliations:** 1Dip. di Meccanica, Matematica e Management, Politecnico di Bari, 70131 Bari, Italy; roberto.spina@poliba.it; 2Istituto Nazionale di Fisica Nucleare (INFN)—Sezione di Bari, 70125 Bari, Italy; 3Consiglio Nazionale delle Ricerche—Istituto di Fotonica e Nanotecnologie (CNR-IFN), 70126 Bari, Italy

**Keywords:** FFF parts, material characterization, mechanical testing

## Abstract

The objective of the present work is to study the influence of color additives used for the polylactic acid (PLA) filament on the final quality of fused filament fabrication (FFF) parts. The main processing parameters of FFF parts were evaluated, identifying the significant correlations between PLA properties and part performance, using a commercial FFF machine. The quality of the products was evaluated in terms of thermo-mechanical properties such as mechanical strength, principal material temperatures, and viscosity. These last properties were characterized using differential scanning calorimetry (DSC) for the thermal measurements and a rotational rheometry (RHEO) for viscosity measurements. Cylindrical specimens were then produced for the compression test. The experimental activity and related testing of products are fully described, pointing out a significant difference in performance between parts made of different colored filaments.

## 1. Introduction

Additive manufacturing (AM) is defined in the ISO/ASTM 52900:2015 standard as the “process of joining materials to make parts from 3D model data, usually layer upon layer, as opposed to subtractive manufacturing and formative manufacturing methodologies” (Lee at al. [1]). New opportunities are nowadays offered by AM processes in terms of innovative production paradigms and manufacturing possibilities to fabricate moderate to mass-produced quantities of individually customized products. A reduction can be achieved in manufacturing lead times, new design developments, and time to market, as well as an increase in meeting customer demand. Small-volume manufacturing in the AM industry is expected to increase because of the use of AM in automobile parts and aerospace engine production. The growing interest in developing AM-based systems over conventional paradigms is related to several advantages over traditional processes such as fabrication of very complex geometries with high precision, maximum material savings, flexibility in design and personal customization (Ngo et al. [2]). Among essential technological requirements that needed to be satisfied is the fabrication of robust end-user products with high strength and long-term stability, and a substantial advance in the knowledge of the part behavior under real working conditions. Mechanical properties must be well-known in advance and included in the early design stage to achieve these goals (Attaran [3]). The dependencies of AM techniques on related technologies such as physical modeling, design tools, computing, and process design represent an essential challenge for both applied and basic research. The major obstacles of fully implementing AM technologies are the size restrictions which lead to product segmentation and assembly, low production time, material heterogeneity, and structural reliability because of the limited choice of available materials, high investment cost of the AM equipment, as well as potential issues associated to standardization and intellectual property (Gao et al. [4]). Moreover, most AM processes use more energy than comparable conventional technologies at the process and machine levels (Rejeski et al. [5]).

Fused filament fabrication (FFF) is one of the most popular AM processes used for realizing thermo-plastic prototypes, tools, and low-volume parts. FFF, initially denominated as fused deposition modeling, is an extrusion-based process in which parts are fabricated by melting polymer-based filaments and depositing molten materials on a horizontal build platform. The slice computation with specific software is needed to generate the extrusion trajectories from a 3D CAD model. polylactic acid (PLA) and acrylonitrile butadiene styrene (ABS), polyimide (PI), polycarbonate (PC), polyamide (PA), polyvinyl alcohol (PVA), high-density polyethylene (HDPE), polyetheretherketone (PEEK), and high impact polystyrene (HIPS) are typical materials used (Kumar et al. [6]). FFF products are mainly used for aesthetic parts and assembly models. However, the main goals are the fabrication of real functional parts by improving the print size, output per hour, and printing speed, as well as the realization of molds and tooling for composite manufacturing (Brenken et al. [7]). One of the main constraints to the process development is related to the material used (thermoplastic polymers) and melting behavior. Feeding (tension and compression) and melting (heating) of a high-quality filament should be characterized by superior mechanical properties and thermal stability (warpage) during and after processing to achieve good products (Dizon et al. [8]). Hence, the study of the thermo-mechanical properties represents an essential subject of interest and research of the FFF process (Popescu et al. [9]).

PLA is an interesting material for rapid prototyping because of its low warpage. Its production costs are higher than petroleum-derived, non-biodegradable materials, and it is naturally brittle. Moreover, deviations in dimensional tolerances of produced parts, in terms of design and geometrical features, can be induced from color pigments added to the PLA filament, causing processed parts to be out of the defined features. All the above features can limit the adoption of this material to several short-term applications (Koh et al. [10]). The sources of deviations could be the differences in the coloring process and color added to the raw polymer (Valerga et al. [11]). During the FFF process, the extruded material concurrently cools the layer being printed and re-heats the previous one. The polymer–polymer interface is consequently above the glass transition temperature or melting temperature in a small-time window. The temperature evolution during processing is thus one of the most important fields to monitor because of its significant spatial and temporal variations (Seppala et al. [12]).

For this reason, in-depth tests were performed to improve better knowledge of material characteristics. An accurate description of the material thermal behavior was crucial to predicting the final performance of FFF parts in real conditions. The glass transition and melting temperatures influenced main processing parameters such as the bed and printing temperatures. The bed temperature should be kept below the glass transition temperature to promote the polymer-bed adhesion. The material should be extruded above its melting temperature to guarantee the necessary fluidity. Low glass transition and melting temperatures allowed PLA to be printed with less heat, sometimes reducing the need for a heated print bed.

The goal of the present study is to investigate the thermo-mechanical behaviors of FFF parts, evaluating PLA suitability for a given application. The influence of PLA material and FFF process parameters on the final performance of printed products was investigated. A commercial FFF machine was employed to realize the FFF products. The quality of these products was assessed in terms of mechanical strength and thermal properties. The experimental activity and related testing of products are fully described.

## 2. Material Characterization

The investigated PLA filaments were 1.75 mm in diameter with a dimensional diameter tolerance of ±25 μm. The main physical and mechanical properties, collected from material data sheets (M3D LLC, Fulton, MO, USA), are reported in Table 1.

A small-scale deviation of the filament diameter value is acceptable for stable results, while a high variation is one of the primary sources of excessive changes in strength and elongation. Several measurements were repeated in different sections using a precision diameter gage to take into consideration the difference in diameter, confirming that the declared diameter tolerance was respected. The products were realized using two different PLAs, named Onyx Black and Crystal Clear. Onyx Black is a black PLA, while Crystal Clear is a transparent PLA. Neither the identity nor the composition of either PLA was exactly specified. The percentage of additives and colorants/pigment ranged between 1–5% and 0–5%, respectively. The minimum values of the physical properties were specified with no information about their evolution over time, and no specifications on the influence of the different color on thermal properties.

The thermal characterization of the materials was conducted using differential scanning calorimetry (DSC) and thermogravimetric analysis (TGA) for the thermal properties, whereas rotational rheometry (RHEO) was used for the viscosity measurements.

### 2.1. DSC and TGA

The identification of the thermal properties was carried out with a DSC 403 F3 Pegasus (Netzsch-Gerätebau GmbH, Selb, Germany), with a Rhodium Furnace (sensor type S) and a DSC head (sensor types E). The thermal cycle, as well as the heating and cooling rates, were precisely defined, as described in Spina et al. [13]. Briefly, there were two consecutive runs with a heating step (from 30 °C to 220 °C), a holding step (10 min) and a cooling step (from 220 to 30 °C). The polymorphic aspect of PLA was investigated by analyzing the melting process throughout heating after non-isothermal crystallization. The first heating cycle provided information on material properties related to the filament extrusion. The second cycle reproduced the material properties after printing, supposing that the residence time in the extruder was not sufficiently long to completely erase the crystal nuclei. The baseline was computed by subtracting the signals of a specimen and an empty crucible. All measurements were carried out in the non-isothermal mode with a heating and cooling rate of 7.5 °C/min, using nitrogen (50 mL/min) as the purge gas. The sample weight of the examined material in a closed aluminum pan was approximately 8 mg. The procedure was applied using both Onyx Black and Crystal Clear filaments. Ten specimens for each material were analyzed. The results of the DSC analysis were the identification of the primary thermal properties such as the glass transition, crystallization, and melting temperatures as well as the whole temperature history, according to ISO 11357:2018 [14], as reported in Figure 1 and Figure 2. The average thermogram results are reported in Table 2, with a temperature variation of ±0.4 °C and a heat of fusion/crystallization variation of ±5 kJ/mole.

In the first heating step, Crystal Clear was characterized by a glass transition temperature of 64.5 °C, a cold crystallization temperature of 127.2 °C and a melting temperature of 153.8 °C. The cold crystallization and melt temperatures of Onyx Black were higher, equal to 155.9 and 172.7 °C, respectively, while no difference was recorded for the glass transition temperature, equal to 64.5 °C. An additional cold crystallization temperature was detected for the Onyx Black at 91.8 °C, probably because of the presence of α and α′ crystal form of PLA (Refaa et al. [15]). The heat of fusion, equal to 183.9 and 796 kJ/mole respectively for Crystal Clear and Onyx Black, revealed a higher heat of crystallization of Onyx Black. The heat of fusion was computed as the area under the DSC signal, expressed in kJ/g, divided by the average PLA molar mass equal to 170 g/mol. The cooling step showed that the crystallization temperature of the Onyx Black was higher (126.5 °C) than that of the Crystal Clear (96.0 °C), with a very significant variation in the heat of crystallization. The crystallization of both materials was complete, and the cold crystallization effect was no longer identified in the second heating cycle. The differences in heat melting peaks continued to remain. No degradation effect occurred in either PLA, despite the different values of the heat of fusion and small reductions in the peak temperatures. The results clearly show that crystallinity was color-dependent, confirming the results of previous research (Wittbrodt and Pearce [16]). 

A Pyris 6 TGA (PerkinElmer, Waltham MA, USA) was used for the investigation in the temperature range between 30 and 800 °C. The thermal cycle consisted of a single run with a heating step from 30 to 800 °C. All measurements were carried out with a heating rate of 10 °C/min using a nitrogen atmosphere with a purge rate of 40 mL/min. The sample weight of the examined material in an open alumina crucible was approximately 60 mg. The procedure was applied to both Onyx Black and Crystal Clear filaments. The thermogravimetric curves shown in Figure 3 are the average values of five repetitions for each material. These curves show a single-stage degradation mechanism for both materials. The samples were fully degraded at 450 °C, with a stable residue of the same magnitude up to 800 °C. These results confirmed the presence of mineral fillers (Cicala et al. [17]).

The temperature at 5% mass loss was very similar, respectively, equal to 336.81 and 336.96 °C for Crystal Clear and Onyx Black PLA. The residue contents at 600 °C were 0.68% and 1.20% for Crystal Clear and Onyx Black PLA, respectively. 

### 2.2. Rheological Measurements

The viscosity at different temperatures was measured with a HAAKE MARS III rotational rheometer (Thermo Fisher Scientific Inc, Waltham, MA, USA). The shielding gas was nitrogen to avoid polymer degradation, and the cooling media was water. The specimen was placed between the two highly polished stainless-steel plates of the shearing device at room temperature, heated to the investigated temperature, and held until it was completely melted. A 25.0 mm diameter disk was realized by slowly lowering the two plates to 0.5 mm. This gap was kept constant during the thermal cycle. Each variation caused by thermal expansion was compensated by automatically moving the upper plate. After melt homogenization, shear was promptly applied. The contact between the polymer and the upper plate was continuously maintained. A two-step experimental protocol was applied. Amplitude sweep tests were performed in the first step to identify a stable linear viscoelastic region of the material. The polymer viscosity was measured with frequency sweep tests during the second step. The frequency sweep tests were carried out with a shear value equal to 5 × 10^−3^ because a stable common linear region was identified for shear values between 10^−3^ and 10^−2^. A frequency sweep is a particularly useful test to determine the viscoelastic properties of a material over time. Cox and Merz [18] identified a strict equivalence between the complex viscosity η*(Ω) measured in an oscillatory frequency sweep and the steady shear viscosity η(dγ/d*t*) measured as a function of shear rate dγ/d*t*. Applying this equivalence, the viscosity of the polymer was then computed. Figure 4 shows the viscosity measurements, made with a shear rate ranging between 6 × 10^−2^ and 2 × 10^2^ s^−1^. Shear rate values lower than 6 × 10^−2^ s^−1^ were not used because the material degraded, as reported by Balani et al. [19].

The viscosity variation field of the Crystal Clear was higher than that of the Onyx Black at all temperatures tested. In particular, the Newtonian plateau of the Crystal Clear ranged between 726 and 4826 Pa s in the investigated temperature interval of 170–220 °C, whereas the Newtonian plateau of Onyx Black ranged between 417 and 3921 Pa s. The differences in the viscosity values became smaller with increasing shear rate. In general, the high filament viscosity led to under-extrusion (impossibility to supply the right amount of material), causing missing layers, thin layers, or layers with random dots and holes. For this reason, a low viscosity is highly desirable, thus avoiding shallow values, causing over-extrusion (excess of deposited material).

The next step was the analysis of the linear viscoelastic *G′* (loss) and *G′′* (storage) moduli. Figure 5 and Figure 6 show the experimental measurements of both materials at shear rates ranging between 6 × 10^−2^ and 2 × 10^2^ s^−1^.

Typical Maxwellian behavior was observed for both PLAs, independent of the presence of the colorant, in which the viscous nature was dominant at low frequencies (*G′′* > *G′*), and the elastic behavior was dominant at high frequencies (*G′′* < *G′*). The crossover frequency was computed to determine the characteristic relaxation time by taking the inverse of the crossover frequency. Figure 7 contains the crossover frequencies of both PLAs as a function of the temperature. The measured crossover frequency was approximately 5–80 s^−1^, corresponding to a characteristic relaxation time of 0.0125–0.2 s.

The crossover frequency of Onyx Back was higher than that of Crystal Clear. As a consequence, the lower viscosity and elasticity of Crystal Clear resulted in a much higher propensity to flow/drip soon after the filament deposition compared with Onyx Black. This behavior had been previously recorded in similar experiments (Cicala at al. [17]). 

Another important material parameter to evaluate was the molecular weight distribution (MWD). Gel permeation chromatography (GPC) is the conventional method to determine the MWD of a polymer (Van Dijk et al. [20]). However, GPC has some disadvantages such as the polymer to be measured must be dissolved in a solvent first, the instrumentation is expensive, the measurement is a time-consuming procedure and, the method is less sensitive for high molecular components. Evaluating the MWD from rheological data is an interesting alternative to GPC by correlating the MWD with the material functions. Inherent in the rheological data is information on the sample modulus and relaxation times, which are significantly affected by molecular entanglements and the molecular weights. The procedure started with the construction of the master curves of the two PLAs and the computation of the relaxation spectra from oscillation frequency sweep data. The MWD was then computed using these data (Shawn and Tuminello [21]). Figure 8 shows the MWD results of the two polymers, pointing out that the difference was very small.

This result pointed out that the colorant additive affected the PLA crystallization, without changing the distribution of the molecular weight. The polydispersity was 1.075 for both PLA polymers.

The analysis of these experimental results revealed that the color additive significantly influenced the thermal and rheological properties of the filaments. According to the above results, a critical analysis of the printing parameters should be performed to optimize the mechanical response of the FFF parts.

## 3. Definition of Process Parameters

∅20 × 20 mm^3^ cylindrical specimens were produced using a Micro + 3D Printer (M3D LLC, Fulton, MO, USA) to be used in the compression tests. The dimensions of the compression specimens were defined according to UNI EN ISO 604 [22]. Several parameters were adjusted on the FFF machine to achieve an accurate print. In this research, the investigated parameters were the infill distance *D*, print temperature *T,* and print speed *S*. The print temperature is the temperature of material extruded from the printhead nozzle, whereas the print speed defines the velocity at which the printhead moves during deposition. The extrusion speed was a function of the print speed to guarantee stable flow conditions during deposition. The modification of the printing conditions caused a change in the extrusion flow and hence changed the shear rate, consequently inducing a variation in the polymer melt viscosity. The specimen was realized as a shell-infill structure, consisting of a solid outer shell and a porous infill. The solid shell was the external wall to preserve the outline shape, whereas the internal infill featured a honeycomb-like mesostructure to maintain the strength of the structure (Fu et al. [22]). The infill density defined the amount of plastic used to fill the honeycomb structure. A higher infill density was associated with more plastic inside the part. The distance between each infill line, defined as the infill distance, was then chosen, having the same effect of changing the infill density. The lowest and the highest possible values of the infill distance were 0.5 mm and 10.0 mm, respectively, 0.5 mm being for a filled part and 10.0 mm for a part with a maximum fill density equal to 4%. Figure 9 shows the internal geometry of components with different infill line distances. A reduction in the infill distance, corresponding to an increase in infill percentage, directly increased the part strength.

The internal structure was built using a linear infill strategy, creating a grid-shaped infill, printed in one diagonal direction per layer. Additional parameters were the bed temperature set to 40 °C and the choice of a brim support type. All the above parameters influenced the thermo-mechanical history of the FFF parts. The selection of the proper range variation was fundamental, as reported by Brenken et al. [7]. During the filament deposition, the hot material wetted, reheated, or re-melted previously laid down and bond material beads. The wetting process determined the contact area between beads, while more prolonged exposure to elevated temperatures promoted the coalescence of adjacent beads and inter-diffusion of polymer chains through the interface. A high temperature encouraged the diffusion process between adjacent beads, whereas too low a temperature caused a decrease in the molecular mobility, hindering the diffusion process. Hence, the temperature history at the bead-to-bead interface, induced by the thermal heat fluxes, was a critical factor for the bond formation. Additionally, the viscosity rapidly increased with the material crystallization, interrupting the bond formation process. The vertical build orientation was chosen for all specimens, considering that the build orientation is usually one of the factors having the most influence on mechanical strength (Chacon et al. [23]) as well as being one of the leading causes of circularity errors and geometrical imperfections (Eswaran et al. [24]). Moreover, the fabrication and measurement were realized in a realistic environmental status. This status referred to uncontrolled printing, storage, and shipping conditions, as well as no specific specimen conditioning (Tymrak et al. [25]).

## 4. Results and Discussion

The product testing was based on uniaxial compression tests to evaluate the material behavior under crushing loads. The initial tests were extended to assess the influence of process parameters on specific performance variables. Scanning electron microscope images were used to support the results.

### 4.1. Compression Test

Uniaxial compression tests were carried out at room temperature to determine the deformation mechanisms and study the static behavior. The stress–strain curves of samples realized with different process parameter sets and materials were analyzed. A servo-controlled 4485 machine (Instron, Norwood, MA, USA) was used to carry out the tests. The load and displacement accuracies were 0.25% with the 200 kN load cell and 2.5 × 10^−5^ mm. The initial compression deformation rate was 4.167 × 10^−3^ 1/s, for a crosshead speed equal to 5.0 mm/min, by initially applying a 0.5 N preload before each test. Each sample was identified by the triples AAA/BBB/CCC in which AAA was the extrusion temperature, BBB the infill distance, and CCC the print speed. The measured stress–strain curves, shown in Figure 10 for the Crystal Clear sample, showed three different zones. The material behavior was linearly elastic between *O* and *P*_1_; a roughly slow stress gradient was detected between *P*_1_ and *P*_2_, and the material finally showed a rapid stress increase between *P*_2_ and *P*_3_. Then the load was released, and the specimen partially returned to its initial height.

The same figure shows the results of Onyx Black specimens, fabricated with the same process parameter set. The elastic modulus *E* and yield strength σ*_Y_* were lower than those of the Crystal Clear, revealing a decrease in the mechanical performance. The stress-strain curves of both specimens were overlapped for strain values higher than 15%. As for the increase in the infill distance, specimens fabricated with the process parameter set 215/5.25/45 exhibited behavior closer to that of cellular materials (Spina [26]). The specimen section consisted of a cluster of rectangular cells. The deformation of some cells became critical, involving the adjacent sections in the collapse. A partially folded specimen was obtained at the end of the test. The specimen underwent a continuous reduction in the stiffness and a gradually weakening response until the maximum stress was reached. The collapse of a section determined a decrease in stress, contributing to an oscillating trend in the stress–strain curves. The specimens made of Crystal Clear again exhibited better mechanical performance than Onyx Black.

### 4.2. Experimental Analysis

A Design of Experiment (DoE) technique was used to plan and analyze the experimental campaign. The investigated process parameters were the infill distance D, print temperature T, and print speed S, the value intervals of which are reported in Table 3.

A Box-Behnken design was used because the treatment combinations were at the midpoints between the edges of the process space and the center, defining a well-known feasible exploration range of the defined process parameter sets (*NIST* [27]). This rotable design used three levels for each factor involved, for a total number of 13 tests for each material with three repetitions for each test. The results are reported in Table 4 in terms of the main response variables such as specimen weight *W*, elastic modulus *E*, and yield strength σ*_Y_* as functions of the input variables infill distance *D*, print temperature *T*, and print speed *S*.

The analysis of variance (ANOVA) of the weight *W*, elastic modulus *E*, and yield strength σ*_Y_* showed that the variable having the most influence was the infill distance *D*, followed by the print temperature *T* and finally print speed *S*. The values of the coefficient of determination *R^2^* of all response variables were higher than 0.75 and 0.96 for both materials using a linear or a quadratic model respectively. These values pointed out an excellent agreement between the predicted and the actual values, confirming the validity of the use of regression models for engineering applications. The ANOVA of the above variables performed with a quadratic model is reported in Table 5 and Table 6, for Crystal Clear and Onyx Black, respectively.

The following analysis was carried out on the regression equations between input and response variables. Figure 11 shows the 3D responses surface plots of specimen weight *W* as a function of the infill distance *D* and print temperature *T*, using a constant value of the print speed *S* for both materials. The red dots are the experimental points, while the surfaces were the predicted responses computed using the regression equations. The infill distance *D* was the factor having the most influence, whereas print temperature *T* had no significant effect on the investigated process response. No significant variation existed between specimen weight using different materials.

Figure 12 shows the elastic modulus *E* as a function of the infill distance *D* and the print temperature *T*. In this case, the print temperature *T* had a slight influence on the investigated process response for both materials. In particular, an increase in the print temperature caused an increase in the elastic modulus, especially for a low infill distance. This effect was more considerable for Onyx Black than Crystal Clear.

Figure 13 shows the yield strength σ*_Y_* as a function of the infill distance *D* and the print temperature *T*. Also, in this case, the print temperature *T* only had a slight influence on the investigated process response for both materials. An increase in the print temperature T had a significant effect on the decrease of the yield strength for Crystal Clear, whereas a moderate variation was detected for Onyx Black.

Based on the above results, the best performances were achieved using a low infill distance with a high print temperature, independently of the print speed in the examined range, for both PLA materials.

### 4.3. Scanning Electron Microscope (SEM) Analysis

The SEM images of various filament threads were acquired using a Gemini SEM 300 (Carl Zeiss AG, Oberkochen, Germany). Samples of both PLA materials were cut from ∅20 × 20 mm^3^ cylindrical specimens with infill distances equal to 0.5 and 2.5 mm. Samples were sputter-coated by depositing single layers of gold in a very controlled thickness pattern and then inserted into the SEM chamber for the observations. Figure 14 shows the acquired SEM images in the center of the specimens with a magnification of 70×.

The images show details of the printing process, such as the shape of the deposited roads, the periodicity, and the voids left between deposited roads. The evaluation was initially performed on the images of 0.5 mm infill distance samples. The roads of Onyx Black were more defined, and gaps between roads were more uniform. On the contrary, the roads of Crystal Clear tended to expand, covering the gaps between adjacent roads in several locations. This behavior was in accordance with the lower melting point that could cause the re-melting of the previously deposited layers. This reduced non-uniform gap distance could be one of the reasons for the higher value of the elastic modulus *E* and yield strength σ*_Y_* than those of Onyx Black. The situation was different for samples with an infill distance of 2.5 mm. The roads made of Crystal Clear were sharper, and overlaps between roads at a different height were clearer. On the contrary, a poor gap road section was observed for Onyx Black, with a very non-uniform section. This behavior was in accordance with the lower value of viscosity than Crystal Clear. Once again, this could be identified as the reason for higher values of *E* and σ*_Y_* of the Crystal Clear specimens. The SEM results clearly showed that the best road morphology was achieved with the Crystal Clear specimen.

## 5. Conclusions

The author has investigated the fabrication of FFF parts using two commercial PLA materials with different colors (natural for Crystal Clear and black for Onyx Black). The filaments were characterized by rotational rheology, showing a significant difference in terms of viscosity and moduli between materials. Several samples were then fabricated with different process parameters such as infill distance, print temperature, and print speed. The specimens were subjected to compression tests. The results showed that the mechanical properties were strictly influenced by the color additive. In particular, products made with Crystal Clear exhibited better performance in terms of elastic modulus and tensile strength than Onyx Black. The SEM analysis of the FFF specimens showed more defined roads and more uniform gaps between roads for the Onyx Black PLA. In future work, the crystallization kinetics of PLA should be numerically computed in order to optimize the printing conditions in the FFF process with more reliable process parameters.

## Figures and Tables

**Figure 1 polymers-11-01984-f001:**
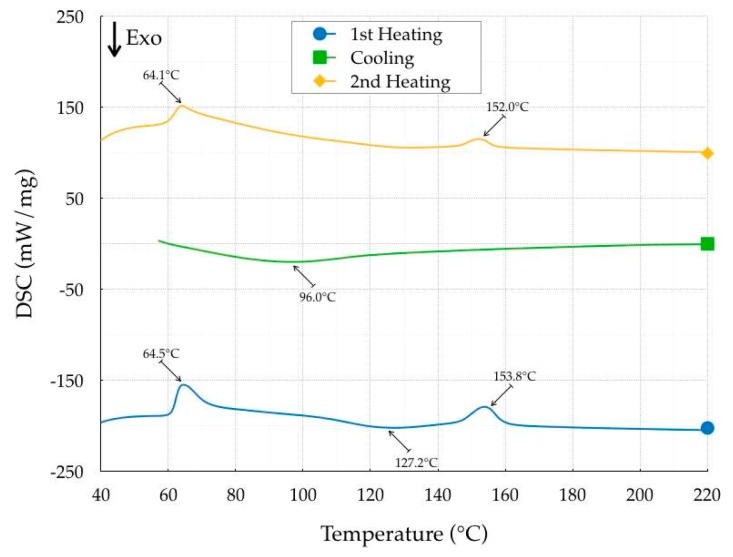
Thermogram of the Crystal Clear PLA.

**Figure 2 polymers-11-01984-f002:**
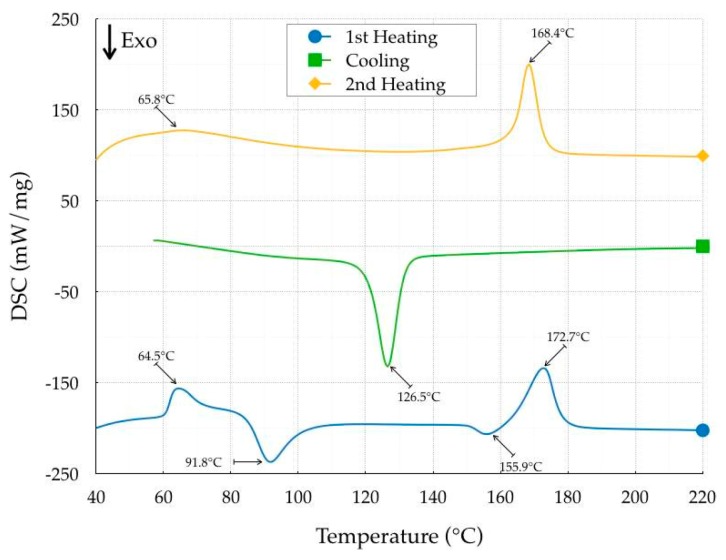
Thermogram of the Onyx Black PLA.

**Figure 3 polymers-11-01984-f003:**
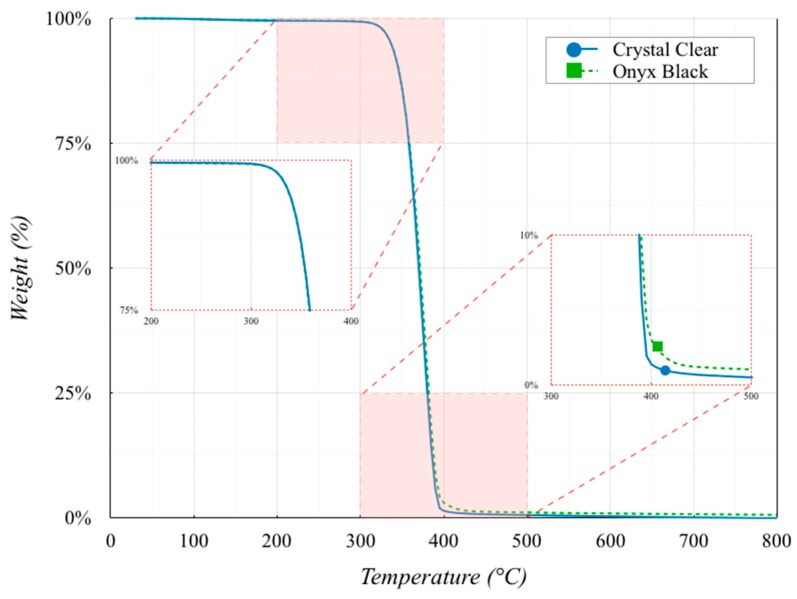
Thermogravimetric curves of the two PLAs.

**Figure 4 polymers-11-01984-f004:**
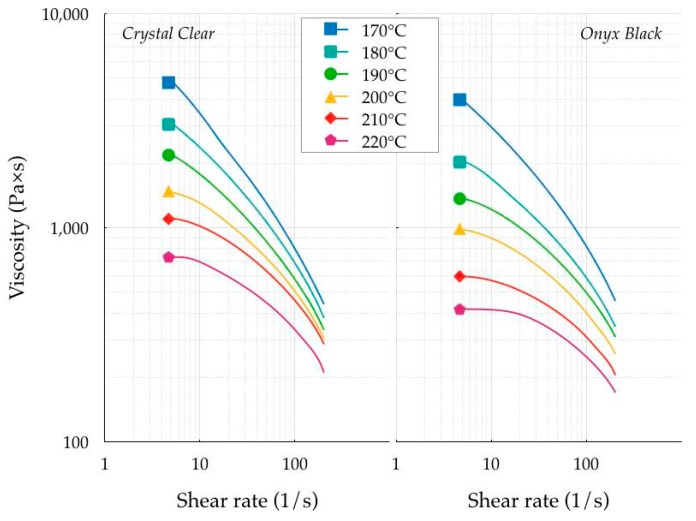
Viscosity measurements.

**Figure 5 polymers-11-01984-f005:**
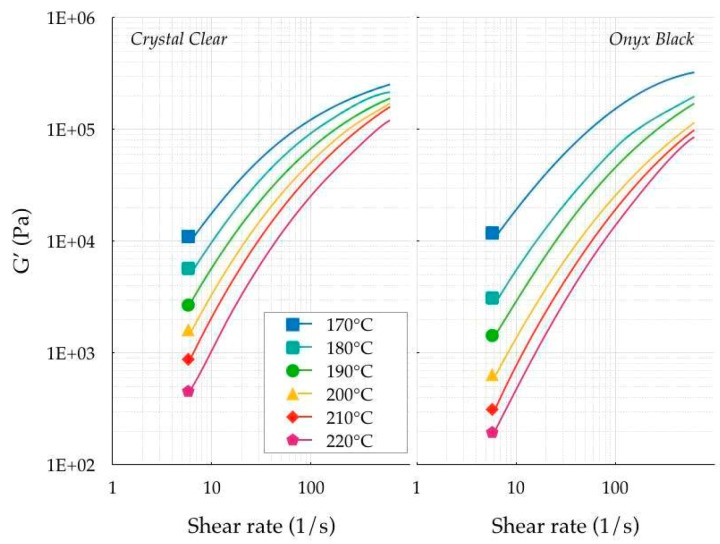
Storage modulus.

**Figure 6 polymers-11-01984-f006:**
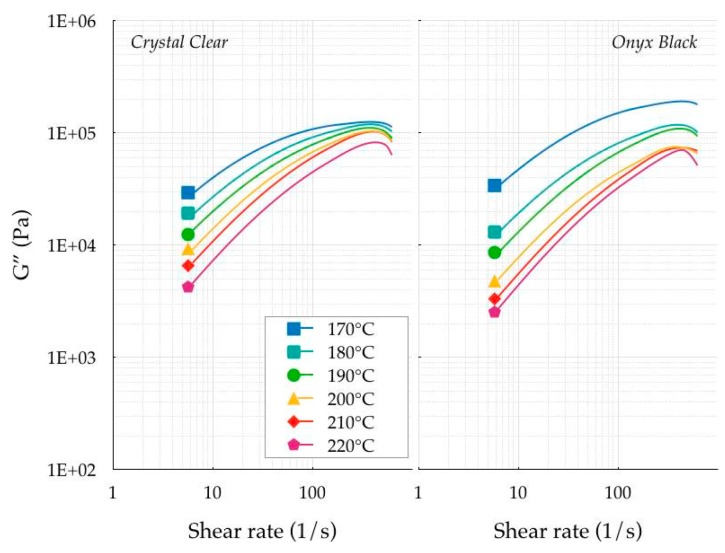
Loss modulus.

**Figure 7 polymers-11-01984-f007:**
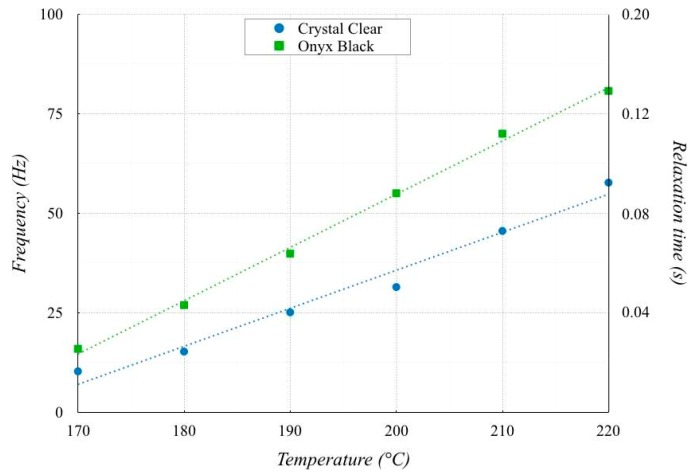
Crossover frequencies.

**Figure 8 polymers-11-01984-f008:**
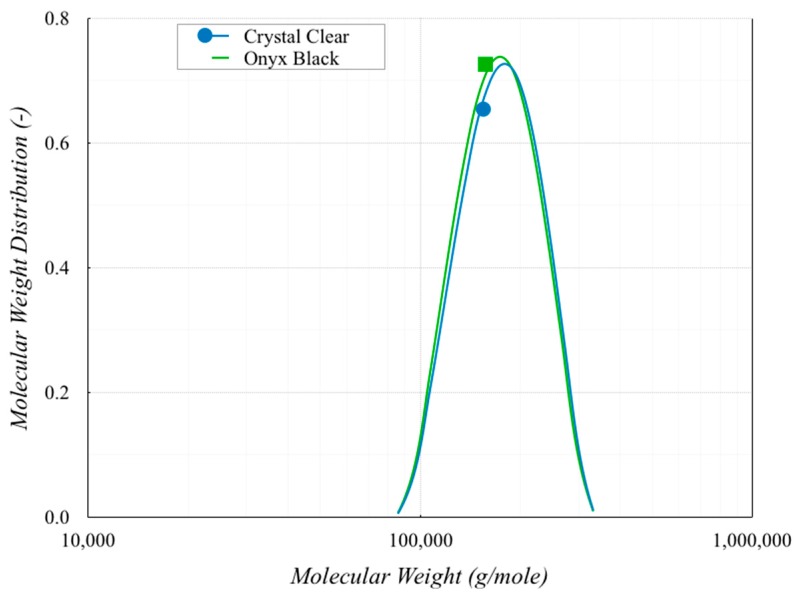
Molecular weight distribution.

**Figure 9 polymers-11-01984-f009:**
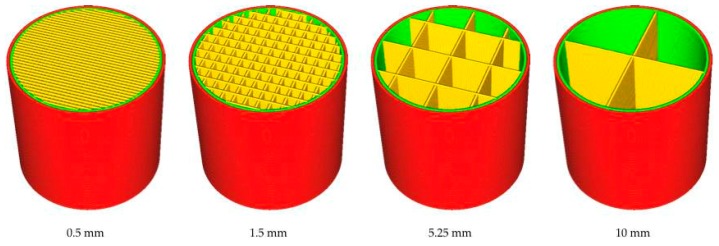
Variation of the infill distance.

**Figure 10 polymers-11-01984-f010:**
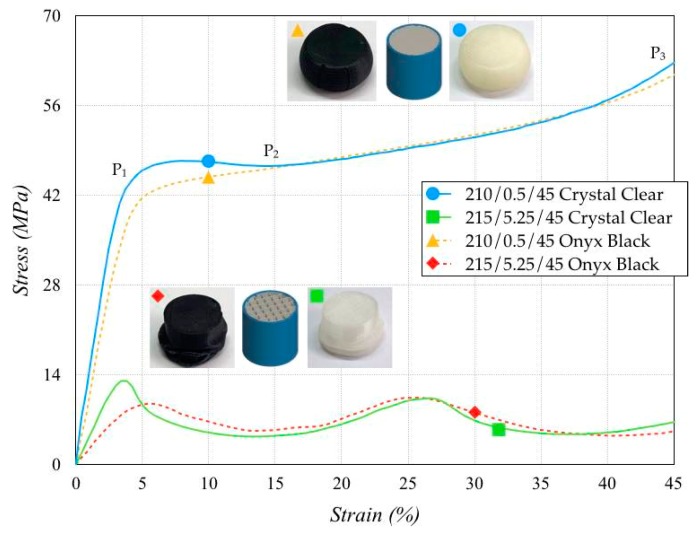
Stress–strain curves.

**Figure 11 polymers-11-01984-f011:**
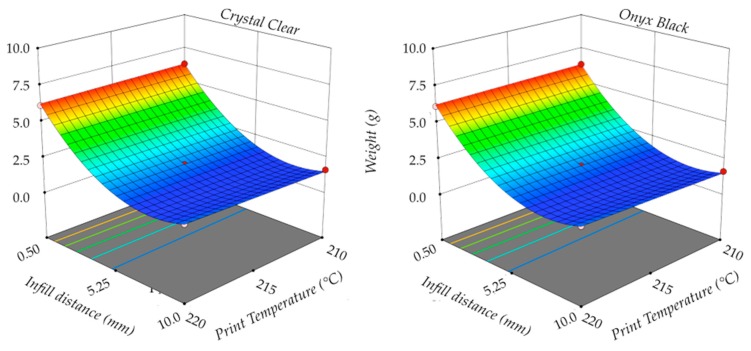
Weight.

**Figure 12 polymers-11-01984-f012:**
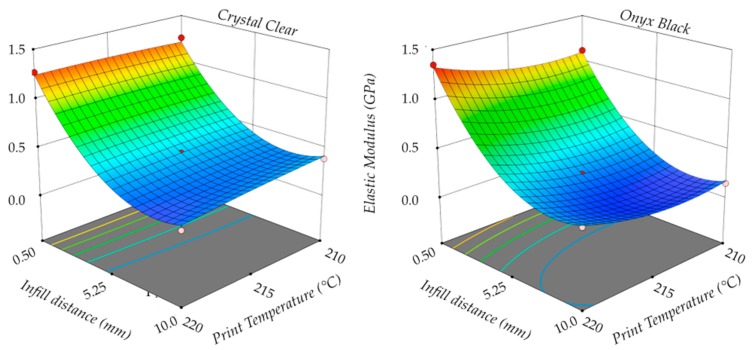
Elastic modulus.

**Figure 13 polymers-11-01984-f013:**
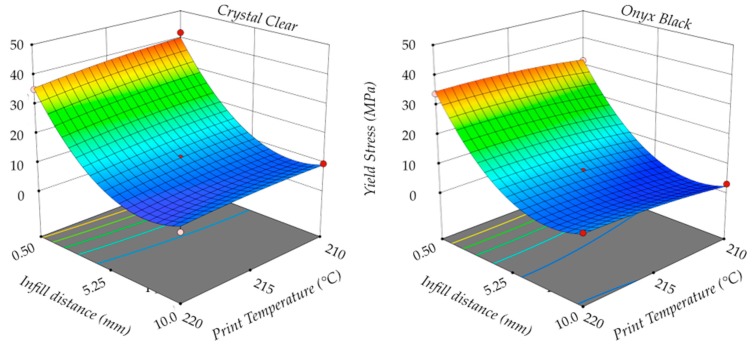
Yield stress.

**Figure 14 polymers-11-01984-f014:**
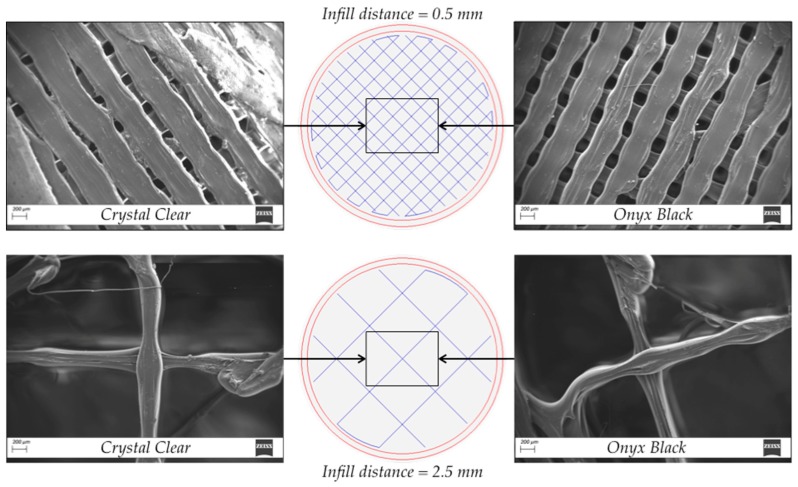
SEM images.

**Table 1 polymers-11-01984-t001:** Physical and mechanical properties of supplied polylactic acids (PLAs).

Properties	Value	Unit
**Physical**		
Density	1.3	g/cm^3^
Glass Transition Temperature	45.0	°C
Melting (Softening Temperature)	160.0	°C
Decomposition Temperature	250.0	°C
**Mechanical**		
Young’s Modules	3.5	GPa
Yield Tensile Strength	25.0	MPa
Ultimate Tensile Strength	38.0	MPa
Elongation at Break	6.0	%
Coefficient of Thermal Expansion	68.0	µm/m/°C

**Table 2 polymers-11-01984-t002:** Differential scanning calorimetry (DSC) results (temperature variation ±0.4 °C, heat of fusion/crystallization variation ±5 kJ/mole).

Properties	Onyx Black	Crystal Clear	Unit
1st Heating			°C
Glass Transition Temperature	64.5	64.5	°C
Cold Crystallization Temperature	155.9	127.2	°C
Melting (Softening) Temperature	172.7	153.8	kJ/mol
Heat of Fusion	796.8	183.9	
Cooling			
Crystallization Temperature	126.5	96.0	°C
Heat of Crystallization	1219.1	282.5	kJ/mol
2nd Heating			
Glass Transition Temperature	65.8	64.1	°C
Cold Crystallization Temperature	-	-	°C
Melting (Softening) Temperature	168.4	152.0	°C
Heat of Fusion	1066.1	94.0	kJ/mol

**Table 3 polymers-11-01984-t003:** Process parameters.

Parameters	Min.	Ave.	Max.	Unit
Variable	210	215	220	°C
Print Temperature *T*	0.50	5.25	10.00	mm
Infill Distance *D*	30	45	60	mm/s
Print Speed *S*				
Fixed				
Bed Temperature		40		°C
Wall Speed	15	23	30	mm/s

**Table 4 polymers-11-01984-t004:** Compression test results.

Crystal Clear		⟵ID^*^⟶					Onyx Black	
*g*	GPa	MPa				*g*	GPa	MPa
W	E	σ_Y_				W	E	σ_Y_
1.878	0.386	9.643	210	/10	/45	1.468	0.145	3.073
2.060	0.426	11.521	220	/5.25	/30	2.130	0.418	7.208
6.277	1.255	42.019	210	/0.5	/45	6.326	1.119	31.789
6.244	1.037	39.510	215	/0.5	/30	6.360	1.043	32.321
2.131	0.427	10.404	210	/5.25	/30	1.896	0.247	4.852
2.082	0.433	11.513	210	/5.25	/60	1.849	0.202	4.035
6.186	1.331	35.773	215	/0.5	/60	6.304	1.086	37.361
2.018	0.402	9.752	220	/5.25	/60	2.152	0.396	9.585
1.862	0.382	9.384	215	/10	/60	1.680	0.170	3.375
2.097	0.448	11.655	215	/5.25	/45	2.022	0.243	7.487
1.827	0.399	10.770	215	/10	/30	1.587	0.099	6.324
6.136	1.271	35.162	220	/0.5	/45	6.137	1.351	33.983
1.744	0.252	6.601	220	/10	/45	1.880	0.304	7.050

^*^ XXX/YYY/ZZZ ➔ XXX Print Temperature (°C) / YYY Infill distance (mm) / ZZZ Print speed (mm/s).

**Table 5 polymers-11-01984-t005:** ANOVA of Crystal Clear.

	Weigth (W)				Elastic Modulus (E)				Yield Strength (σ_Y_)			
Source	SS	MS	F-value	p-value	SS	MS	F-value	p-value	SS	MS	F-value	p-value
Model	50.05	5.56	4994	< 0.0001	1.95	0.22	28.17	0.0096	2201.68	244.63	42.15	0.0053
A—Temperature *T*	0.02	0.02	18.87	0.0225	0.00	0.00	0.36	0.5885	13.89	13.89	2.39	0.2196
B—Distance *D*	38.42	38.42	34,505	< 0.0001	1.51	1.51	195.80	0.0008	1683.91	1683.91	290.11	0.0004
C—Speed *S*	0.00	0.00	1.46	0.3136	0.01	0.01	1.09	0.3736	4.18	4.18	0.72	0.4584
A × B	0.00	0.00	0.01	0.9231	0.01	0.01	0.73	0.4558	3.64	3.64	0.63	0.4863
A × C	0.00	0.00	0.01	0.9231	0.00	0.00	0.03	0.8752	2.07	2.07	0.36	0.5924
B × C	0.00	0.00	1.94	0.2578	0.02	0.02	3.14	0.1747	1.38	1.38	0.24	0.6590
A^2^	0.00	0.00	1.05	0.3808	0.00	0.00	0.04	0.8603	1.06	1.06	0.18	0.6983
B^2^	8.55	8.55	7681	< 0.0001	0.29	0.29	37.18	0.0089	350.40	350.40	60.37	0.0044
C^2^	0.00	0.00	0.01	0.9459	0.00	0.00	0.07	0.8144	0.07	0.07	0.01	0.9185
Residuals	0.00	0.00	-	-	0.02	0.01	-	-	17.41	5.80	-	-
Total	50.05	-	-	-	1.98	-	-	-	2219.09	-	-	-

SS ➔ Sum of Squares, MS ➔ Mean square.

**Table 6 polymers-11-01984-t006:** ANOVA of Onyx Black.

	Weigth (W)				Elastic Modulus (E)				Yield Strength (σ_Y_)			
Source	SS	MS	F-value	p-value	SS	MS	F-value	p-value	SS	MS	F-value	p-value
Model	54.25	6.03	346	0.0002	2.40	0.27	107.85	0.0013	2221.75	246.86	268.38	0.0003
A—Temperature *T*	0.05	0.05	2.61	0.2048	0.07	0.07	28.89	0.0126	24.77	24.77	26.93	0.0139
B—Distance *D*	42.11	42.11	2416	< 0.0001	1.88	1.88	761.25	0.0001	1671.34	1671.34	1817.03	< 0.0001
C—Speed *S*	0.00	0.00	0.00	0.9764	0.00	0.00	0.11	0.7603	1.67	1.67	1.81	0.2710
A × B	0.05	0.05	2.83	0.1913	0.00	0.00	0.54	0.5161	0.79	0.79	0.86	0.4212
A × C	0.00	0.00	0.07	0.8108	0.00	0.00	0.05	0.8320	2.55	2.55	2.77	0.1945
B × C	0.01	0.01	0.32	0.6120	0.00	0.00	0.08	0.7966	15.96	15.96	17.35	0.0252
A^2^	0.00	0.00	0.00	0.9748	0.02	0.02	9.52	0.0539	2.15	2.15	2.33	0.2240
B^2^	8.90	8.90	510	0.0002	0.34	0.34	137.16	0.0013	354.63	354.63	385.54	0.0003
C^2^	0.00	0.00	0.02	0.8973	0.00	0.00	0.76	0.4464	0.02	0.02	0.02	0.8873
Residuals	0.05	0.02	-	-	0.01	0.00	-	-	2.76	0.92	-	-
Total	54.30	-	-	-	2.41	-	-	-	2224.51	-	-	-

SS ➔ Sum of Squares, MS ➔ Mean square

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
