# Peer review of "Performance Analysis of Colored PLA Products with a Fused Filament Fabrication Process"

_polymers, 2019, doi:10.3390/polym11121984_

Round 1

Reviewer 1 Report

Dear Authors,

I have carefully read your draft paper and concluded that your study is useful and interesting and may be acceptable for publication after some major revisions. Also, I have a few questions and concerns with your work as presented, which I invite the authors to address or explain, and which are detailed below.

Manuscript Comments and Questions:
- Moderate English changes required. Even though the grammar of this work appears correct in most cases, I think the paper can be clearer and easier to read. The use of shorter sentences can help to understand them.

Introduction:
The introduction provides sufficient information about the content. However, the abstract provides little information about the work. Try to highlight a little better the novelty of your scientific study, even to advance some result.

Experiments:
Some details about the experimental methodology carried out are missing:
- Figure 7 does not provide information.
- How many measurements per sample have been made? and the tolerances?
- Have any standards been followed?
- How many identical samples have been inspected (for DSC analysis)?
- How many similar cylindrical specimens were produced (with the same parameters)? It does not have a minimum number of identical samples (the standard indicates 5) and a dispersion in results. This is not sufficient to present firm results.

In some cases it does provide data such as the specimens, the equipment or the standard used. Review and complete this data throughout the methodology.

Results and Disscussion:
The results should be reinforced with further trials. The rapidity of the additive manufacturing means that more samples can be made in a relatively short time. It should include dispersed results and develop better conclusions.

Please note that the comments are intended merely to assist the authors in improving the paper and ensuring that published papers are of the highest quality. They are in NO WAY intended to discourage or demean the authors personally

Sincerely,

Author Response

Manuscript Comments and Questions:

Q. - Moderate English changes required. Even though the grammar of this work appears correct in most cases, I think the paper can be clearer and easier to read. The use of shorter sentences can help to understand them.
R. English was checked and corrected.

Introduction:

Q. The introduction provides sufficient information about the content. However, the abstract provides little information about the work. Try to highlight a little better the novelty of your scientific study, even to advance some result.
R. The abstract was extended and objective made clearer.

Experiments:
Some details about the experimental methodology carried out are missing:

Q. - Figure 7 does not provide information.
R. Figure 7 was removed from the manuscript.

Q. - How many measurements per sample have been made? and the tolerances?
R. Figure 7 was removed from the manuscript.

Q. - Have any standards been followed?
R. Dimensions of the compression specimens were defined according to ISO 604:2008. DSC results were extracted according to ISO 11357:2018.

Q. - How many identical samples have been inspected (for DSC analysis)?
R. 10 specimen per each material.

- How many similar cylindrical specimens were produced (with the same parameters)? It does not have a minimum number of identical samples (the standard indicates 5) and dispersion in results. This is not sufficient to present firm results.
R. Three repetitions for each test were realized, after producing ten specimens for each test. In this way, stable processing conditions were achieved.

Q. In some cases it does provide data such as the specimens, the equipment or the standard used. Review and complete this data throughout the methodology.
R. information was added to the text.

Results and Discussion:
Q. The results should be reinforced with further trials. The rapidity of the additive manufacturing means that more samples can be made in a relatively short time. It should include dispersed results and develop better conclusions.
R. Results were revised and additional information added.

Please note that the comments are intended merely to assist the authors in improving the paper and ensuring that published papers are of the highest quality. They are in NO WAY intended to discourage or demean the authors personally
R. Thanks for your support and precious suggestions.

Reviewer 2 Report

In this manuscript, Spinastudied the fabrication of FFF parts by using Crystal Clear and Onyx Black based commercially available PLA materials. The resulting filaments were evaluated by rotational rheology which exhibited dramatically different viscosity and moduli between two materials. Moreover, the specimens were assessed by compressive tests, which demonstrated the products made with Crystal Clear had enhanced performance of elastic modulus than the black one. Overall, the manuscript is well written. The data is thorough to support the conclusions. The research is quite interesting and indicated an important role of coloradditives in material properties for additive manufacturing. In view of this, I would like to recommend acceptance of this manuscript to Polymers. However, the author should address my comments below.

Comments

Did the author characterize the molecular weights and polydispersity of those two PLA materials? MW and PDI are very important parameters for the physiochemical properties of polymers. It will be great to compare those informations for Crystal Clear and Onxy Black.

Author Response

Q. Did the author characterize the molecular weights and polydispersity of those two PLA materials?
MW and PDI are very important parameters for the physiochemical properties of polymers.
It will be great to compare those informations for Crystal Clear and Onxy Black.

R. The analysis of the MWD and PDI was added to the text.

Reviewer 3 Report

The article by Spina on the effects of colouring PLA filaments for FDM printing is extremely timely and very well written and of considerable interest; as such there some minor and one major points to be addressed.

Major point; from the text it appears that unprinted/extruded filament is used. Between first and second DSC cycles there are considerable differences these should be considered in the context of the second heating cycle representing the printed object and a third cycle should be carried out. At the same time TGA experiments need to be undertaken - these will show if the change in melting is associated with the loss of some material - probably water or solvent. This will allow deeper discussion.

Minor Points

Table 1 is assembled from various sources but does not take into account the large differences in PLA properties which arise between different manufacturers and even patches from the same source in my hands these can be up to 10% for batches and over 10% between sources. Some care should be expressed on the numbers in Table 1

The author states numerous times that the properties are colour dependent - this is both true and untrue - in fact the differences arise from the dye/colourant used and how it integrates into the filaments and interacts with polymer molecules. Please make this clear, as other producers may use different dyes it is easy to conceive of two "black" filaments having widely differing properties! The same is true in a different way for the transparent filament - natural PLA is offwhite and not transparent so this filament has been untreated.

Apart from the above this paper is excellent and after correction can be published as a priority paper

Author Response

Major point
Q. From the text it appears that unprinted/extruded filament is used. Between first and second DSC cycles there are considerable differences these should be considered in the context of the second heating cycle representing the printed object and a third cycle should be carried out.
R. The analysis was carried-out in a previous paper by the same author (DOI: 10.1063/1.5034987).

Q. At the same time TGA experiments need to be undertaken - these will show if the change in melting is associated with the loss of some material - probably water or solvent. This will allow deeper discussion.
R. The TGA results were added to the manuscript.

Minor Points
Q. Table 1 is assembled from various sources but does not take into account the large differences in PLA properties which arise between different manufacturers and even patches from the same source in my hands these can be up to 10% for batches and over 10% between sources. Some care should be expressed on the numbers in Table 1.
R. Variations were specified in the text and table.

Q. The author states numerous times that the properties are colour dependent - this is both true and untrue - in fact the differences arise from the dye/colourant used and how it integrates into the filaments and interacts with polymer molecules. Please make this clear, as other producers may use different dyes it is easy to conceive of two "black" filaments having widely differing properties! The same is true in a different way for the transparent filament - natural PLA is offwhite and not transparent so this filament has been untreated.
R. The information was added to the text

Apart from the above this paper is excellent and after correction can be published as a priority paper
R. Thanks for your support and precious suggestions.

Reviewer 4 Report

There are too many errors in the manuscript and the paper was badly organized. 

Author Response

Q. Extensive editing of English language and style require.

R. Manuscript was improved and corrections performed by a native speaker reader lecturer.

Round 2

Reviewer 1 Report

Dear author,

You have corrected all the comments indicated, expanding the content and discussing the results better. I am satisfied with the corrections made, although there still seems to be some flaw in the language.
I have recommended to the editor its publication in the journal after a last revision of the language.

Kind regards,

The reviewer

Author Response

Q. I have recommended to the editor its publication in the journal after a last revision of the language.
R. The English language was further improved. All corrections were done

Reviewer 4 Report

The Figure captions should be provided in detail. The grammar needs to be extensively checked since there are too many errors throughout the text.  I wonder how the author plotted the Figure 8. By using the GPC data or rheology data. The author should point out the significance of present work.

Author Response

Q. The Figure captions should be provided in detail.
R. The figure captions were improved.

Q. The grammar needs to be extensively checked since there are too many errors throughout the text.
R. The English language was further improved. All corrections were done

Q. I wonder how the author plotted the Figure 8. By using the GPC data or rheology data.
R. The reference to the work of MT Shawn and WH Tuminello was added. This paper explains the relation between viscosity and MWD.

Q. The author should point out the significance of present work.
R. The importance of the work was explained in the abstract and conclusions. The article investigates the color influence on thermo-mechanical properties of FFF products.

Round 3

Reviewer 4 Report

None